# Climate variability on Fit for 55 European power systems

**Matteo De Felice**[1], **Derck Koolen**[1,2,3] *, **Konstantinos Kanellopoulos**[1], **Sebastian Busch**[1], **Andreas Zucker**[2]

**1** European Commission, Joint Research Centre, Petten, The Netherlands, **2** European Commission, Directorate-General for Energy, Brussels, Belgium, **3** Utrecht University School of Economics, Utrecht, The Netherlands

* derck.koolen@ec.europa.eu

**Data Availability Statement:** Input data and the model outputs are available in the following two Zenodo repositories: 1. https://doi.org/10.5281/zenodo.7065568 2. https://doi.org/10.5281/zenodo.7996208.

## Abstract

The use of variable renewable energy sources to generate electricity introduces a dependency on meteorological factors into power systems. With the renewables share growing globally, often driven by political pressures, the reliability and efficiency of power systems are increasingly affected by this dependency. In this paper, we investigate the impact of the natural variability of meteorological parameters on the European power system in 2030. We specifically focus on (1) analysing the main European weather patterns affecting renewable energy production and (2) understanding the co-variability of this production among European countries. The identification of a set of patterns in the behaviour of key power system operation indicators allows us to analyse the relationship between large-scale weather regimes and daily power system operations in a 2030 European energy context. Regarding renewable generation, analysis of the co-variability shows that European power systems tend to form two clusters, in each of which all the regions tend to show a positive correlation among themselves and a negative correlation with the other cluster. Our analysis of the most important large-scale weather regimes shows that during cyclonic patterns, the carbon intensity of all the European power systems is lower than normal, while the opposite happens during blocking regimes.

## 1. Introduction

With the demand for sustainable energy on the rise, numerous governments and policy makers around the world are announcing pronounced ambitions to increase the share of renewable energy production in the electricity generation mix. In the European Union (EU), as part of the Fit for 55 package of proposals for delivering the European Green Deal, the Renewable energy directive puts forward a legal framework for the development of renewable energy across all sectors of the EU economy. The directive has been revised numerous times to accelerate the roll-out of renewables and reflect increasingly ambitious climate targets. In 2023, the EU agreed on a revised binding renewable target of 42.5% of gross final energy consumption [1].

**Funding:** The author(s) received no specific funding for this work.

**Competing interests:** The authors have declared that no competing interests exist.

In the last twenty years, the European continent has seen a remarkable increase in renewable energy generation. The total capacity of renewable electricity generation in Europe went from 221 GW (with 13 GW of solar/wind) in 2000 to 725 GW (with 427 GW of wind/solar) in 2021 (+230%) [2]. With most renewable production being difficult to predict by nature, dealing with the variability of renewable energy sources (RES) is still one of the biggest challenges for efficiently integrating low-carbon sources in power systems [3]. Since RES operations are tightly linked to weather (due to the availability of wind, sunshine or precipitation), understanding the impact of meteorological and climatic variables on electricity systems is becoming increasingly important for multiple stakeholders along the value chain, from utilities providing (ancillary) services to policy makers ensuring security of supply [4]. In general, introducing inter-annual variability into power system models can lead to more robust estimations of the storage capacity needed in systems with a high penetration of renewables [5]. Furthermore, assessing and quantifying the variability of electricity generation and related makes it possible to improve the design of low-carbon energy systems and increase the penetration of RES with power system reliability in mind.

There exists a growing stream of literature assessing the impact of climate variability and change on electricity systems and markets [4, 6]. Most articles, however, reflect on the impact of a specific climatic variable on one specific electricity system parameter. For example, previous studies analysing the impact of RES variability on European electricity systems have mostly focused specifically on wind and solar resources [7–10] or power system planning [5, 11–13]. Moreover, the majority do so by focusing on a specific country or region in an isolated manner, for example by analysing the sensitivity of the power sector to climate change (for example for different global locations [14–16]. Our paper aims to provide a more comprehensive assessment by analysing in an integrated future European energy context the impact of climate variability on power system operations and indicators, including generation from renewable energy sources (solar, wind and hydropower), demand, carbon dioxide intensity and cross-border flows.

The work addresses two different aspects: 1) co-variability among European countries and 2) analysis of the predominant weather-based patterns. The co-variability of power production from renewable sources in different areas has usually been studied with a focus on the complementarity between wind and solar [17–21]. Similar assessments extended to power systems operations are, however, not common [22]. Regarding weather patterns, we link the operations of European power systems to a set of specific large-scale weather conditions, using the patterns identified in [23] and applied in successive works [24–27]. While recent work has investigated the impact of weather patterns at a large scale [28–30], our analysis provides insights into the impacts of frequent patterns in a future European energy context. By associating the identified weather patterns with a configuration of the European power system, in terms of renewable energy generation, cross-border exchanges and other parameters, the analysis allows us to investigate the relationship between large-scale weather regimes and daily power system operations in a 2030 European energy context. As such, the work enables policy-makers to assess with efficiency the impact of climate variability in decarbonising electricity systems.

Below, we elaborate on the data and models used to perform our modelling analysis. Next, we discuss the impact of natural variability of meteorological parameters from three different perspectives: 1) measuring power system variability; 2) measuring the co-variability between regions and; 3) identifying patterns in the impact of weather regimes. We conclude by discussing the implications of our work for power system planners and policy makers.

## 2. Data & models

This paper investigates the impact of climate variability on wind, solar, hydropower generation and electricity demand in 34 inter-connected European power systems using a power system model based on the European Commission MIX 2030 scenario [31]. The MIX scenario is one of the European Commission's core policy scenarios from the Fit for 55 package [32]. It was produced as a common tool for analysis in the impact assessments of various initiatives in the European Green Deal policy package. This scenario is the central policy scenario, and it assumes an extended Emissions Trading System (ETS) in road transport and buildings, medium to high ambition for energy efficiency and renewable energy policies. It delivers emission reductions of 55% compared to 1990, reducing 2030 energy demand by around 9% compared to the REF 2020 scenario and reaching a share of renewables of 38.4%.

Climate variability is modelled using weather-driven inputs in the power system modeling (Table 1). The selected inputs are the most significant factors in explaining how weather affects supply and demand: they include all the most important renewable energy sources and the electricity demand. Hydropower plays–both today and in the EU 2030 scenarios–a critical role in providing the seasonal storage of electricity. For an extended discussion about the aspects of power systems affected by climate we refer to [33–35].

The dataset we use for solar, wind and hydropower modelling is the Pan-European Climate Database v3.1 (PECD) created by ENTSO-E (European Network of Transmission System Operators for Electricity) and used in the European Resource Adequacy Assessment 2021 (ERAA 2021). The dataset consists of a statistical downscaling from the ERA5 atmospheric reanalysis [37] for the following energy variables: electricity demand (not used in this study), wind (on- and off-shore) and solar capacity factors, hydropower inflows and run-of-river generation.

The electricity demand data we used has been built following the methodology described in ([31], Section 5.5). The methodology is based on the demand for all the European countries, disaggregating heating & cooling (assumed to be little affected by meteorological conditions). Similarly to the PECD dataset, the electricity demand uses meteorological variables from the ERA5 atmospheric reanalysis.

The results shown in this study are based on 34 climate years: each climate year consists of a set of inputs for solar, wind, hydropower generation and electricity demand based on the historical meteorological data for the specific year. A climate year is a physically consistent set of synthetic time-series used as input for the power system model for solar generation, wind generation, hydropower generation and electricity demand. Basically, a climate year represents a what-if scenario, assuming the same installed capacity and technology. The use of climate years can be considered the best way to address the uncertainty derived from meteorological conditions. For example, the work by Collins et al. [10] uses climate years between 1985 and 2014 to analyse the inter-annual variability across Europe. Our approach covers not only wind and solar but also hydropower and demand, increasing the possibility to explain the complex

**Table 1. Source data for the weather-driven inputs used in the power system modelling.**

| Input | Unit measure | Original temporal resolution | Data source |
|---|---|---|---|
| Solar (PV) power | Capacity factor (%) | hourly | [36] |
| Wind power (on- and off-shore) | Capacity factor (%) | Hourly | [36] |
| Hydropower inflow | Capacity factor (%) | Weekly | [36] |
| Run-of-river | Capacity factor (%) | Daily | [36] |
| Electricity demand | Hourly demand (MW) | Hourly | [31] |

interactions between climatic conditions and power systems operations. The use of climate information derived from ERA5 atmospheric reanalysis makes it possible to introduce interannual variability consistently for all the regions analysed [37]. We analyse the interannual variability of the selected inputs in the S1 File.

The simulations in this study are conducted using the METIS power system model [38, 39]. While the model is based on proprietary software, all the data that we have used–as well as the results–are freely available with an open license. The METIS model can simulate the hourly operations of energy assets over a year, minimising the system cost maintaining the supply/demand in each node. Each node of the model represents a country and the interconnector between countries can exchange electricity according to their net transfer capacity. METIS solves a linear economic dispatch problem including all the generation technologies specified in the MIX scenario (as described in [31]). More details on the METIS modelling framework can be found in [38–40].

The MIX scenario was originally developed using PRIMES (Price-Induced Market Equilibrium System) [41]. The scenario covers all the EU plus the United Kingdom and is focused on the long-term energy transition including technology changes and policy measures. The model based on METIS, used in this article, was created following the methodology documented in [31].

Model inputs and outputs are available online with open license [42, 43].

## 3. Results

Our simulations generate results on the operations of power system models for 34 countries using 34 different climatic years (1982–2015). To enhance readability, we group the countries in seven regions: Central Eastern Europe (CEE), Central Western Europe (CWE), Iberia, Italy, Northern, South Eastern Europe (SEE) and UK & Ireland (Fig 1). This classification, as visualized in Fig 1, aims to group together the countries with similar characteristics, and follows the approach used in the "Gas and electricity market reports" published by the European Commission.

To characterise power system operations, we use the following six indicators calculated at hourly level:

1. $CO_2$ intensity: the amount of $CO_2$ emissions emitted per MWh of electricity generated ($gCO_2$eq/kWh).

2. Inflow: the quantity of water that a hydropower plant can use to be stored or converted into electricity

3. Net export: the annual difference between cross-border exports of electricity and imports.

4. Peak net demand: the peak demand of electricity minus the variable renewable generation

5. Solar generation: the amount of electricity generated by solar power in a year

6. Wind generation: the amount of electricity generated by wind power in a year

The chosen metrics were selected to provide an overview of the power systems' general characteristics, rather than using more detailed metrics (e.g. as in [10]) that would inevitably increase uncertainty due to the challenge of characterising electricity dispatching and power markets dynamics in future policy scenarios. Below, we discuss the impact of the natural variability of meteorological parameters on these parameters considering three different perspectives: (1) measuring power system variability, (2) measuring the co-variability between regions and (3) identifying patterns in the impact of weather regimes.

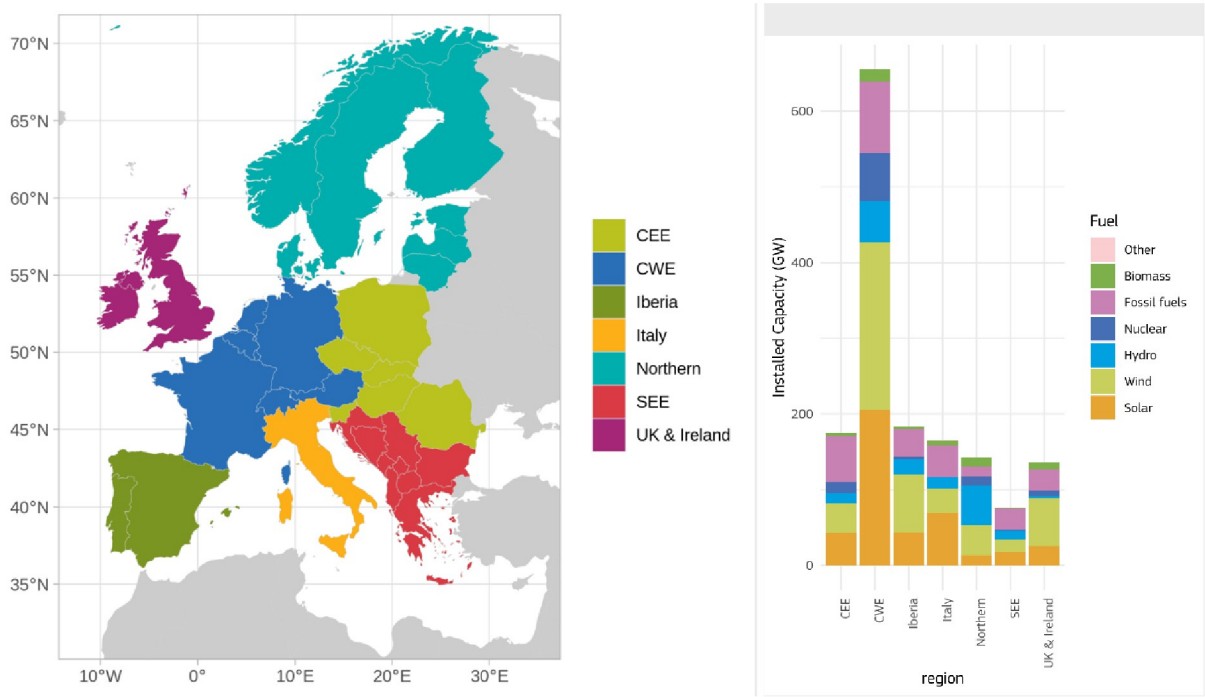

**Fig 1.** Left: Countries modelled in the simulation and the grouping used for the analysis. Right: installed electrical capacity for each region according to the MIX 2030 scenario.

### 3.1 Power system variability

We use the coefficient of variation (CV) to measure the year-on-year variability. The CV (also called relative standard deviation) is defined as the ratio between the standard deviation ($\sigma$) and the average ($\mu$): $CV = \sigma/\mu$.

Table 2 shows the interannual variability, measured using the CV expressed as a percentage, for all the selected metrics.

The table shows that among all the renewables, hydropower seems to be the most variable source, which is consistent with recent statistics [44]. The hydropower inflow in the Iberian region shows a very high variability (as investigated for Spain in [45]) with a clear correlation with its net electricity export ($r_s$ 0.72 CI 95% [0.51, 0.85]), highlighting the capability of the Iberian region to provide low-carbon electricity to its neighbours (i.e., France, in the analysed

**Table 2. CV (expressed as a percentage) of the year-over-year variation of the annual value for the selected variables.** Shaded cells represent the highest value for each variable.

| Region | CO$_2$ intensity | Hydropower Inflow | Net export | Net demand | Solar gen. | Wind gen. |
|---|---|---|---|---|---|---|
| CEE | 7.1% | 13% | 64.9% | 1.8% | 3.2% | 5.3% |
| CWE | 10.7% | 8.1% | 15.7% | 5% | 3.2% | 4.9% |
| Iberia | 13.6% | 32.6% | 84% | 6.7% | 2.2% | 5.3% |
| Italy | 2% | 11.9% | 15.4% | 2.2% | 2.5% | 5.5% |
| Northern | 17.5% | 10.6% | 51.4% | 2.2% | 3.2% | 5.4% |
| SEE | 5.3% | 13.5% | 33.9% | 1.3% | 2.5% | 3.5% |
| UK & Ireland | 15.2% | - | 24.3% | 5.9% | 3% | 5.7% |

scenario). Beyond hydropower, other renewable energy sources like wind and solar show relatively low variability, as the annual capacity factors for these sources can be predicted with relatively high precision [8].

We further measure the relationship, as expressed with a linear regression, between the penetration of renewable electricity sources and emission intensity (Fig 2). This analysis indicates how the variability of a specific source of electricity can affect the emissions of the regional power systems. As expected, we observe a strong relationship between the share of renewable energy sources in a region's electricity generation and the $CO_2$ intensity of that electricity. The magnitude of the effect is stronger for variable renewable energy sources wind and solar than for hydropower, related to the relative (in)flexibility of the generation source and its position in the merit order curve.

The results are in line with [10], who focus on the analysis of the variability of European power systems from a similar perspective, especially regarding the variability of $CO_2$ intensity and wind and solar generation. We observe two key areas of divergence in the literature affecting these power system variables: 1) the use of different scenarios (for example the European Commission MIX 2030 versus the ENTSO-E visions scenarios from the TYNDP 2016 in [10]) and 2) differing approaches to the variability of renewable energy sources (for example the lack of hydropower in [10] introducing major differences in the supply/demand balance in hydropower-rich regions.

## 3.2 Co-variability

We define co-variability as how the change in one region relates to changes in the rest of the continent. Co-variability happens mainly for two reasons: 1) European power systems are inter-connected and can exchange electricity and 2) the meteorological phenomena influencing the power systems (e.g., wind lulls) are often large-scale, large enough to affect multiple regions (if not the entire continent) at the same time. The co-variability between two variables is measured using the Spearman rank correlation coefficient and, where necessary, a confidence interval of the value is calculated using a Fisher z-transformation, following the implementation provided by [46]. We analyse the co-variability starting from the year-over-year

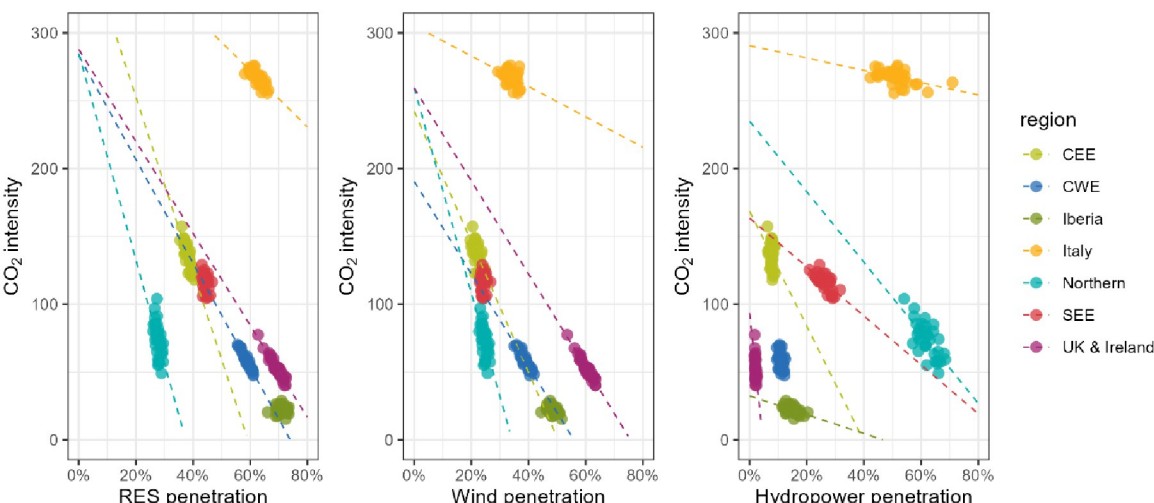

**Fig 2. Linear regression showing the relationship between penetration of electricity sources and the $CO_2$ intensity (measured in $gCO_2$/kWh).** Only groups with linear regression p<0.001 are shown.

variations of wind and solar generation (Fig 3). We observe how the correlation in the northern regions (mostly wind-driven) shows a cluster, albeit less evident during summer. The southern regions (Iberia, Italy and SEE) are weakly correlated with each other and show a negative correlation with the northern regions. The strongest negative correlation is between Iberia and Northern, -0.49 (CI 95% [-0.84, -0.13]) during winter, and between CEE and Iberia, -0.45 (CI 95% [-0.78, -0.12]) throughout the year.

This first step offers a glimpse of the spatial relationships among the European regions regarding the characteristics of their power systems. As a next step, we explore in more detail the relationship between the other aspects of the simulated power systems (Fig 4).

Solar and wind generation show correlation across multiple zones, suggesting that the meteorological drivers substantially affecting solar and wind are large-scale. This is consistent with Fig 3 as well as with other results found in scientific literature [47, 48]. For wind generation, we can observe negative correlations between the two clusters: for example, Iberian and Northern show a correlation of -0.53 (CI 95% [-0.83, -0.22]), Italy and Northern -0.50 (CI 95% [-0.84, -0.16]) and Italy and UK & Ireland -0.47 (CI 95% [-0.77, -0.16]). We further observe that solar power also shows a positive correlation among multiple regions but it is, in general, weaker than wind.

In terms of other indicators, emission intensity is positively linked in many regions, in particular in UK & Ireland and CWE ($r_s = 0.87$, CI 95% [0.78, 0.95]) and CWE with CEE ($r_s = 0.90$, CI 95% [0.83, 0.98]). Indeed, with emission intensity directly connected to the availability of low carbon/renewables sources, the availability of renewables is correlated across most of the regions. Situations with higher/lower RES generation thus lead to lower/higher emissions and can span multiple regions, if not the entire continent. Finally, the panel on net export (i.e., the difference between export and import at annual level) shows two things:

a) Some regions (often neighbours) can simultaneously experience an increase/decrease of electricity exports, meaning that their supply tends to vary year-over-year with a similar pattern

b) The negative correlation can be seen as a natural implication of electricity trading, but it may also suggest that when a region tends to have a deficit of low-cost electricity another one has a surplus and vice versa.

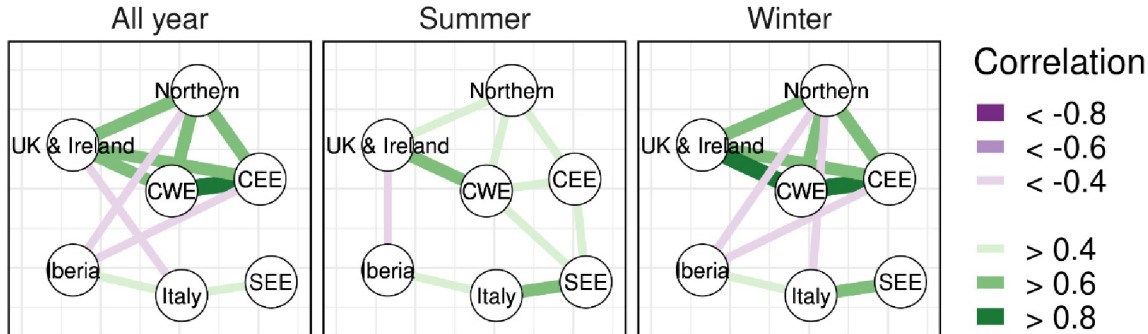

**Fig 3. Year-over-year Spearman correlation of wind and solar generation for the entire year or for summer/winter.** The colours and size of the edges between the nodes is based on the value of the Spearman correlation coefficient. We show only values above 0.4 and below -0.4.

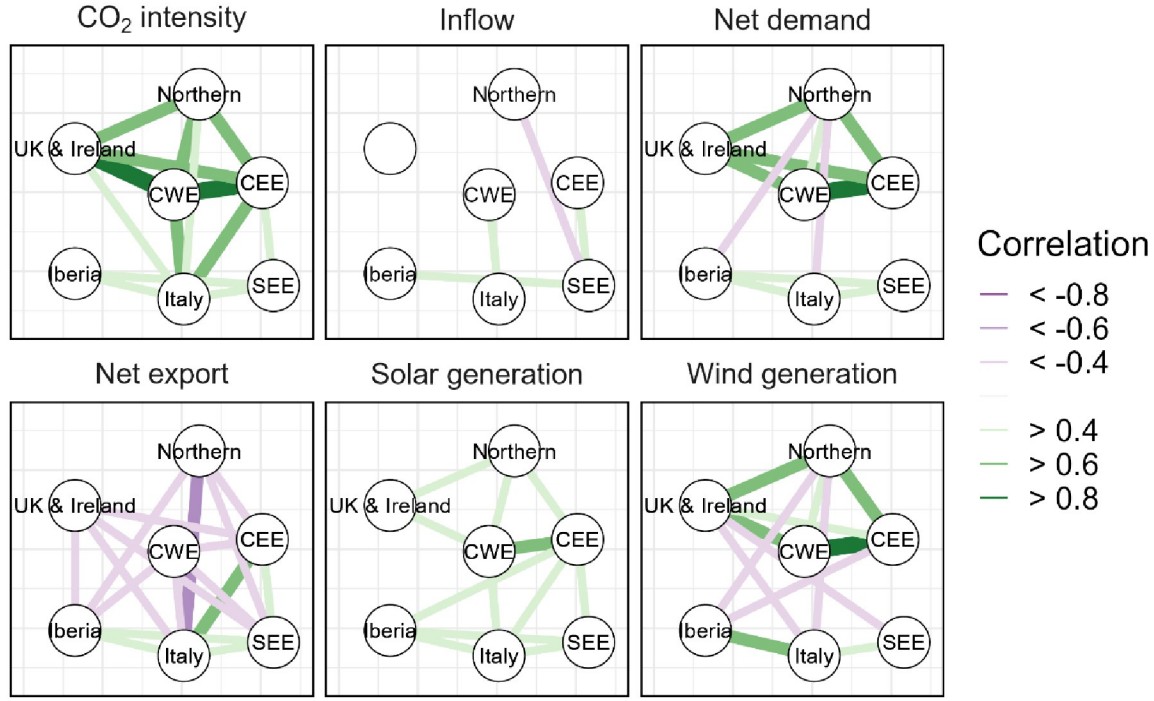

**Fig 4. Year-over-year Spearman correlation for six variables considering the entire year.** The colours and size of the edges between the nodes is based on the value of the Spearman correlation coefficient. We show only values above 0.4 and below -0.4.

### 3.3 Patterns

Weather phenomena affecting European power systems can have a large-scale impact on power systems, i.e., simultaneously affecting multiple countries or even the whole continent. These phenomena can be recurrent and persistent, with similar patterns leading to similar impacts on power systems. A way to assess those patterns is through the analysis of weather regimes. Weather regimes are a classification of recurrent weather states and have been used effectively in many applications, including the analysis of energy systems [23, 30, 49].

In this work, we use a classification introduced by [23], to study the deployment of wind energy in Europe and applied in consequent works, as for example in [24, 27]. We use the patterns identified by Grams for the first time for the analysis of power systems operations; previous work based on this classification has focused mostly on wind.

Fig 5 shows the seven different weather states over Europe in this classification, computed through a clustering algorithm on the observed meteorological daily data for the period 1979–2015. While this classification has been identified for a winter period, it can be associated with year-round phenomena [23]. The regimes are calculated using the geopotential height anomaly on ERA-Interim atmospheric reanalysis and make it possible to characterise large-scale weather conditions that might affect renewable energy generation and power systems operations in general. Three of the seven identified regimes can be associated with the so-called cyclonic-type regime: Atlantic trough (AT), Zonal regime (ZO) and Scandinavian trough (ScTr). A cyclonic regime is often associated with windy conditions in northern Europe and mild temperatures in general. The remaining four weather regimes are 'blocking' regimes: Atlantic Ridge (AR), European blocking (EuBL). Scandinavian Blocking (ScBL) and Greenland Blocking (GL). The blocking regimes normally bring calmer winds, and colder

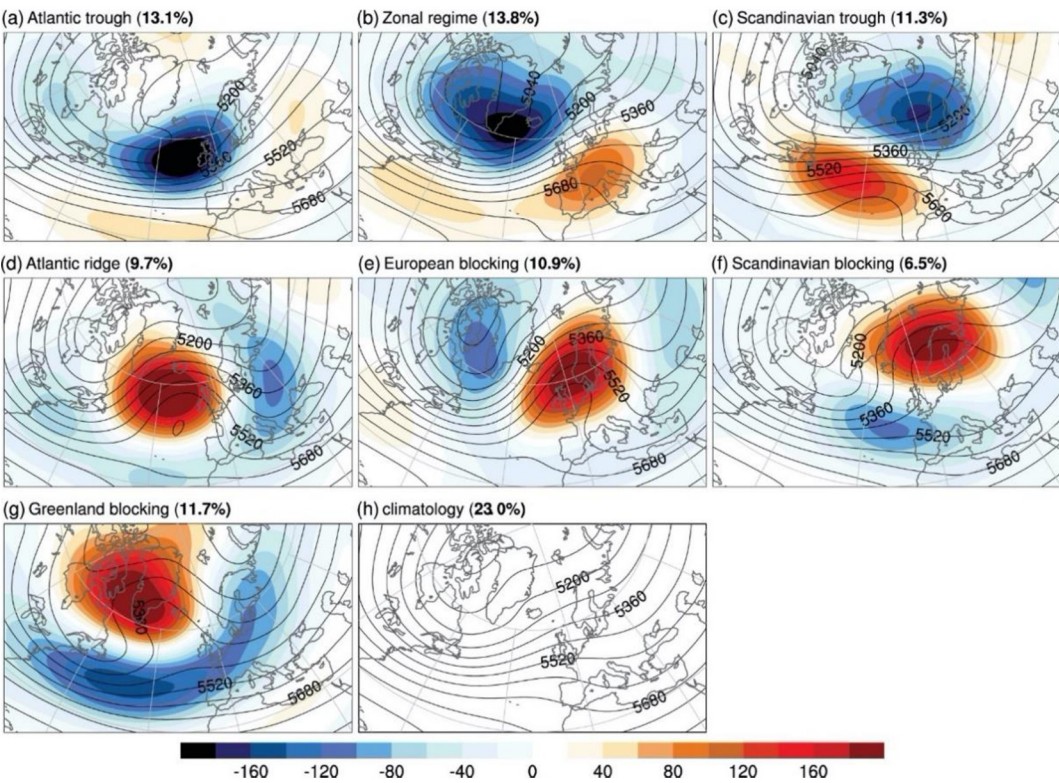

**Fig 5. The weather regimes used in this study introduced in (Grams et al., 2017).** The composite shows the mean 500-hPa geopotential height anomaly in the winter months. The eighth panel represents the 'climatology', the lack of other regimes. The percentage near the name of the regime represents the relative frequency of the regimes. Figure taken from Fig 1 in [24], published with CC BY 4.0 license.

temperatures prevail. A different pattern is associated with each day, allowing us to investigate the average deviation of the analysed variables from their normal values.

For each of the six indicators we calculate the average daily deviation during a specific pattern in relation to the overall mean (Fig 6). In other words, for each of the variables we calculate the percentage difference at day $t$ from the average value (i.e., climatology) for all the days $t$ across all the considered years. We then apply a seven-day moving average to each variable for the calculation of the mean, in order to estimate a more robust climatology.

The change of $CO_2$ intensity shows that during patterns AT, ScTr, Ar, and EuBL the low-carbon generation across Europe seems to follow the same trend: an increase for all the regions during the cyclonic regimes (first two) and a decrease for the rest. The emission intensity is clearly linked with the availability of wind power, as clearly visible in the patterns AT and ScTR where an increased availability of wind is associated with lower emissions across the entire continent. The same reversed pattern is visible in EuBL.

Consistently with the results in [23], we find opposing behaviour for wind power between the northern regions (Northern, CWE, CEE, UK & Ireland) and southern areas (Italy and Iberia) for most of the weather patterns, in particular between the ZO and GL patterns. Indeed, wind and solar power consistently take up opposing positions across most of the patterns, although the variability of solar power is clearly weaker than that of wind.

Some of the patterns are associated with the North Atlantic Oscillation (NAO). The NAO is the most important European weather phenomenon and plays a very important role in

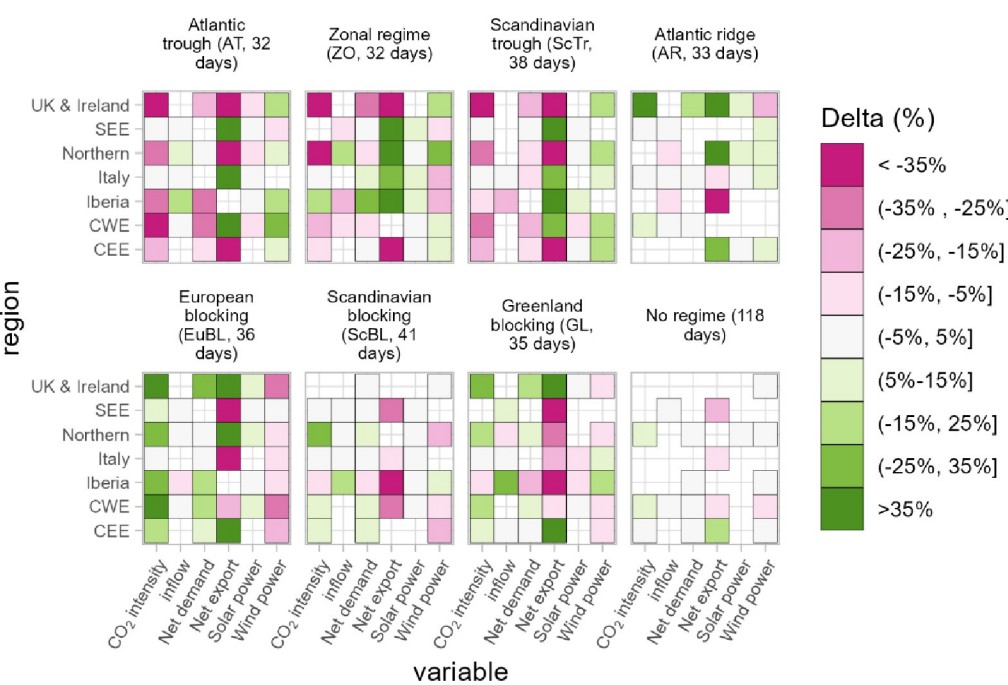

**Fig 6. Percentage change from the daily average across all the climatic years of six energy variables for each pattern considered.** Only the values with the p-value of a T-Test smaller than 0.001 are shown. For each pattern we specify its average number of days per year.

shaping the pattern of energy-relevant variables across the continent (for an in-depth analysis, we refer the reader to [50–55]). In the patterns that can be associated with a positive state of the NAO—as the Atlantic Trough (AT), Zonal Regime (ZO) and Scandinavian trough (ScTr)– there is an increase of wind power in the north and a decrease in the south. Conversely, for the pattern Greenland Blocking (GL), that can be associated with a negative state of NAO, Italy and Iberia show an increase in wind power of between 5% and 15%, with a decrease in the rest of the continent.

We further observe an anti-correlation between variable renewable energy generation and regions associated with the various weather patterns. For example, in the region CWE (428 GW of installed wind and solar), the daily generation of wind and solar is 54 GWh more than the average during the AT pattern, and -37 GWh during the EuBL pattern (Fig 7). The difference between the two patterns, chosen as the most distant in terms of renewable generation, is less marked in the other regions. Moreover, as also mentioned in [23], the figure highlights that the EuBL/AT patterns do not seem to have any impact on wind and solar generation in the SEE region. This is in contrast to the CWE region. In general, as shown in Fig 6, wind generation in SEE seems to exhibit behaviour that does not correlate with the other regions.

For hydropower inflow, the regions tend to show the same sign of change in general, with the notable exception of the pattern EuBL, where CEE, SEE and Iberia shows an increase (particularly large in Iberia >35%) and Northern and UK & Ireland a decrease. A reversed pattern, albeit with a weaker signal, can be observed for ZO.

Analysis of the association between large-scale weather regimes and daily power systems operations shows how multiple patterns can affect the entire continent. Our results regarding the wind and solar power generation in each analysed pattern are consistent with the results

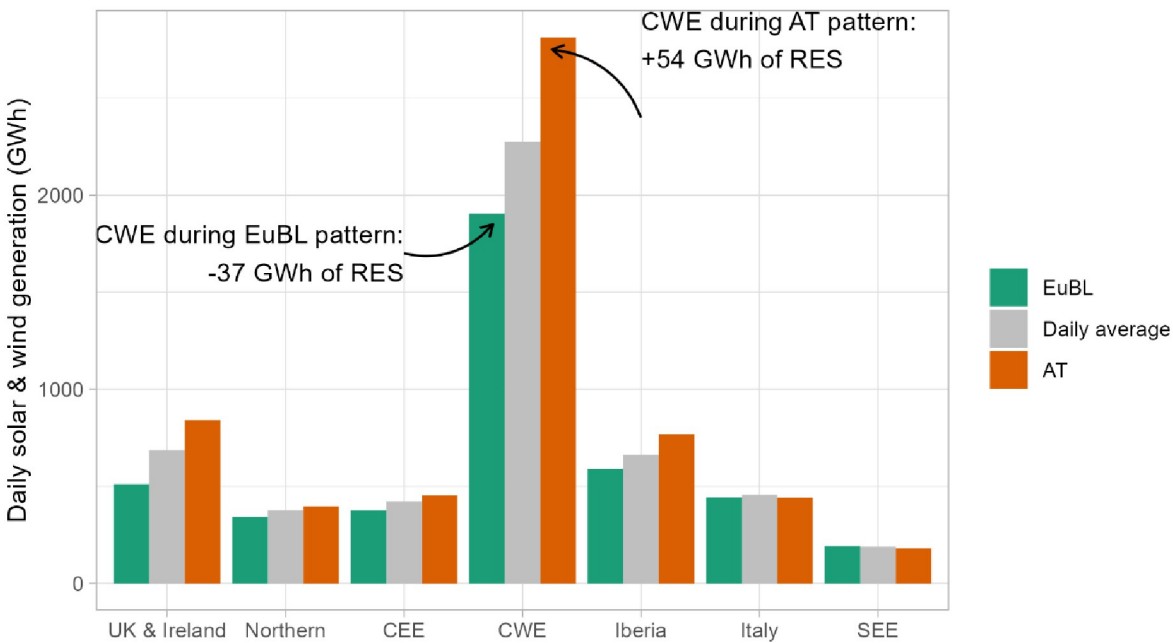

**Fig 7. Difference (in GWh) of daily and wind generation across Europe during the patterns Atlantic Trough (AT, orange) and European Blocking (EuBL, green) for the European regions.**

shown in [23] (for wind) and in [30] (for both wind and solar). In three of the identified patterns (AT, ScTR, EuBL), all the European regions show the same sign of daily generation of wind and/or solar while in the other three patterns, northern and southern regions exhibit opposing behaviours.

During cyclonic patterns, the carbon intensity of electricity is generally lower across the whole continent, while the opposite happens during blocking regimes. This is partially due to the importance of wind in providing carbon-free electricity to the continent, which is mostly installed in central-western countries which tend to have stronger winds during cyclonic regimes [23].

## 4. Discussion and conclusions

This study analyses the impact of year-on-year variability of weather and large-scale weather patterns on the 2030 European power system as envisioned in the MIX scenario of the Fit for 55 policy package of the European Commission. The scenario assumes that 75% of the total installed power generation capacity and 65% of all electricity generated in the EU comes from renewable sources. In electricity systems with such a high dependence on variable renewable generation, characterising the variability is important for several reasons:

1. To evaluate possible seasonal supply/demand scenarios and ensure the efficient operation of seasonal storage levels for electricity generation (mostly hydropower reservoirs in the scenarios analysed).

2. To assess the importance of climatic variability for power system planning purposes. As discussed in [56], when using a single year, the capacity expansion optimisation may be highly dependent on the chosen climatic conditions.

3. To assess power system dynamics during the most prevalent weather patterns in order to consistently link an event (e.g., reduced availability of renewable resources) in one region with the same event in other regions. Furthermore, the results suggest the possibility of using weather patterns to predict power systems' variables using seasonal/sub-seasonal forecasts in Europe as already explored by [57, 58].

Our findings are, as such, of relevance to various audiences involved in the analysis of the European power system, ranging from researchers, to energy system planners and policymakers.

First, for energy system planners, for example for system operators, power generation companies or national regulatory authorities, the climate variability and co-variability are essential to consider for the robust configuration and dimensioning of the power system. To provide for system adequacy, planning decisions need to incorporate variability at the system level over an appropriate time. Moreover, the patterns of co-variability shown in this study offer opportunities for complementarities in system planning that could be realised as synergies through network planning or could result in enhanced revenue and risk hedging opportunities for market participants.

Indeed, we find power systems operations between two or more European regions to be largely correlated. While co-variability is, in general, stronger between neighbouring regions, we also find an evident correlation between some distant regions: for example, the Iberian peninsula and the Northern region exhibit a negative correlation for wind generation, net demand and net export of electricity. Such co-variability is important for system planners as its impact on the design of resilient energy systems may be fundamental to improving the integration of renewable energy sources into European power systems [23].

Our results are among the first to show the comprehensive effect of weather patterns on European power system dynamics. While the results for wind generation are consistent with [23], our findings also create insights into cross-border electricity exchanges and the electricity demand. The results thereby show how the use of power system models enables the quantification of the daily variability of the main energy variables, including aspects of importance for energy system planning purposes (e.g., curtailment and storage operations). Second, for policymakers, the results offer insights into the potential impact on power systems of the transition towards clean primary and mostly variable renewable energy supply. This calls for anticipatory policy and regulatory design as the increasing variability will accompany changes in capacity utilisation and hence also the economics of power system assets. For example, in the recently adopted electricity market design reformation, the European Commission recommends enhanced flexibility in future European power systems in order to accommodate variability in demand and supply patterns, as these are highly dependent on climatic variables [1]. This study is therefore intended to engage policymakers in order to ensure that the impact of climate variability on power system security is adequately assessed.

We find coupling between regions to be at the same time potentially beneficial and a threat to system adequacy and security of supply. For example, the negative correlation of hydropower generation between two regions means that one of the two regions can potentially supply the flexibility needed by the other region with lower levels of water in its hydropower reservoirs. On the other hand, a positive correlation of renewable sources (e.g., wind power) might mean that low-wind periods are potentially extended to multiple regions, leading to large-scale supply/demand issues.

Climatic conditions are not the only source of uncertainty but, especially in systems with large shares of renewable energy generation, they are a key contributor. Variability of production and demand play an important role in the methodology for the short-term and seasonal

adequacy assessment described in the EU directive, depicting a common approach for all EU states for electricity crisis prevention [59]. Finally, we note that the presented projection of climate variability to future scenarios does not consider the effects of climate change, which is known to have a significant impact on European power systems In further analysis, we will consider future changes in atmospheric circulation that may have an impact on weather patterns, leading to possible changes in frequency and persistence [60].

## Supporting information

**S1 File.**
(DOCX)

## Acknowledgments

The authors want to thank Kostas Kavvadias for his support in setting up the modelling framework, Christian Grams for sharing the weather patterns data and Catriona Black for her careful editing.

## Author Contributions

**Conceptualization:** Matteo De Felice.

**Formal analysis:** Matteo De Felice, Konstantinos Kanellopoulos, Sebastian Busch.

**Investigation:** Matteo De Felice.

**Methodology:** Matteo De Felice, Derck Koolen, Konstantinos Kanellopoulos, Sebastian Busch.

**Software:** Matteo De Felice.

**Supervision:** Matteo De Felice, Derck Koolen, Konstantinos Kanellopoulos, Andreas Zucker.

**Validation:** Matteo De Felice, Sebastian Busch, Andreas Zucker.

**Visualization:** Matteo De Felice.

**Writing – original draft:** Matteo De Felice.

**Writing – review & editing:** Matteo De Felice, Derck Koolen, Konstantinos Kanellopoulos, Sebastian Busch, Andreas Zucker.

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
