## [Decision Letter · Decision Letter 0]

12 Apr 2023

PONE-D-23-00558Climate variability on Fit for 55 European power systemsPLOS ONE

Dear Dr. De Felice,

Thank you for submitting your manuscript to PLOS ONE. After careful consideration, we feel that it has merit but does not fully meet PLOS ONE’s publication criteria as it currently stands. Therefore, we invite you to submit a revised version of the manuscript that addresses the points raised during the review process.

 Abstract should provide the most important resultsLiterature review and comparison should be improved to put the work in context

Model description needs improvement

Description of datasets should be more detailed and the presentation of the dataset should be improved. At least a descriptive analysis but taking into account that the data is evolving over time is needed for the paper being published.

Comparison to existing work and working out the differences due to the new data 

We look forward to receiving your revised manuscript.

Kind regards,

Bruno Merk

Academic Editor

PLOS ONE

Journal Requirements:

3. Please ensure that you refer to Figure 4 in your text as, if accepted, production will need this reference to link the reader to the figure.

4. We note that Figures 1 and 4 in your submission contain map images which may be copyrighted. All PLOS content is published under the Creative Commons Attribution License (CC BY 4.0), which means that the manuscript, images, and Supporting Information files will be freely available online, and any third party is permitted to access, download, copy, distribute, and use these materials in any way, even commercially, with proper attribution. For these reasons, we cannot publish previously copyrighted maps or satellite images created using proprietary data, such as Google software (Google Maps, Street View, and Earth). For more information, see our copyright guidelines: http://journals.plos.org/plosone/s/licenses-and-copyright.

 a. You may seek permission from the original copyright holder of Figures 1 and 4 to publish the content specifically under the CC BY 4.0 license. 

Additional Editor Comments:

please follow the recommendations of the reviewers in detail

Reviewers' comments:

Reviewer's Responses to Questions

**Comments to the Author**

1. Is the manuscript technically sound, and do the data support the conclusions?

Reviewer #1: Partly

Reviewer #2: Yes

2. Has the statistical analysis been performed appropriately and rigorously?

Reviewer #1: I Don't Know

Reviewer #2: No

3. Have the authors made all data underlying the findings in their manuscript fully available?

Reviewer #1: No

Reviewer #2: No

4. Is the manuscript presented in an intelligible fashion and written in standard English?

Reviewer #1: No

Reviewer #2: Yes

5. Review Comments to the Author

Reviewer #1: This paper covers an analysis of how climate year variability impacts power system optimization models. The researchers describe the variability of some model outcomes when the input data (renewable generation, and load) is based on different climatic years. It also describes how distinct regions correlate with each other in this regard. Finally, the impact of several known weather patterns on power system operations is described.

As a general remark, much greater attention should be paid to situating this work within the existing literature.

Note: page numbers of article start from 7, not 1.

Major comments:

1. Overall it seems like the paper was written somewhat hastily, without sufficient proofreading. Please improve.

2. Abstract presents only motivation and methodology, please also present major findings and key messages or at the very least suggestions for the possible impact or use of this work and why (as you do on p.20 – 21).

3. The description of the contribution of this work could also be made more clear in the beginning of the paper.

4. p.8, l41 - Collins et al 2018 did not investigate climate variability in a planning context! They use a dispatch (= operational model).

1. Please compare your results to those of Collins et al 2018.

5. More generally, the insights gained from the analysis of weather patterns are not distinguished from the existing literature: What are the lessons learned here, that were not yet found in other works?

6. p. 5, performance metrics

1. Please compare your performance metrics to those of Collins et al 2018.

2. Peak load is very uninformative for adequacy / reliability purposes. Peak net load (load – RES generation) or peak load net of power injection (load – generation + storage + demand response dispatch) would be more informative.

7. Could you comment on how this variability impacts member states ability to reach targets?

8. Model description is lacking. Please provide a brief but clear description of the model that is being used: temporal resolution, power flow approximation, technologies included, ..

9. Datasets can be described better. Data quality is of utmost importance for this type of analysis, yet no information is given on how the climatic variability has been correctly integrated in the datasets that have been used. I suggest to create an extra section in the paper that focuses on Data and Model description.

1. p.8, Results – this includes description of the data and performance metrics, which should be in a separate Methodology section. Similarly some sections of the Introduction section could be moved there.

Minor comments:

1. p.7, l24-25 – how much of that capacity is variable RES (as opposed to e.g. biomass)?

2. p.8 l.77 – please mention that the data is open source here (if I understood correctly).

3. The introduction states that the work covers two different aspects: 1) the co-variability among the European countries and 2) the identification of the predominant weather-based patterns. But then later in the paper, it is mentioned that the weather patterns have been taken from another manuscript, and this paper merely describes their influence on power system operation

4. Could you elaborate on the parameters that you have chosen, why these and not others? E.g. curtailment, load shedding, line usage rates, ..

Formatting:

1. Paragraphs are very short, sometimes just a single sentence. This makes for unnecessarily difficult reading.

2. Some very grammatically incorrect sentences e.g. “The change of CO2 intensity shows that during with the patterns AT, ScTr, Ar and EuBL the low-carbon generation across Europe seems following the same trend:” Please have a native English speaker check the paper.

Reviewer #2: Althouhg I find the topic interesting I think that given taht it seems you are referring to a long time span of years, just a descriptive statistics is not enough to describe the dataset. I would need to see the time evolution plots of the variables under study, so as to see at least if these series are stationary or not. Additionally you mention that the data is available but (maybe it was my fault) but I couldn't find it.

At least a descriptive analysis but taking into account that the data is evolving over time is needed for the paper being published.

6. PLOS authors have the option to publish the peer review history of their article (what does this mean?). If published, this will include your full peer review and any attached files.

Reviewer #1: No

Reviewer #2: No

---

## [Author Response · Author response to Decision Letter 0]

6 Jun 2023

Dear editor,

Thank you for the effort that you put into reviewing our manuscript. We sincerely appreciate the invitation to address the concerns and resubmit our work to PLOS ONE. We have addressed the comments of the reviewers and believe this further improved the quality of our paper. In the attached document, we address the concerns in detail in commenting in a point-by-point manner.

Regarding the style, we would like also to clarify that:

1. We have edited the document to match the PLOS ONE’s style requirements, changing references and cross-references

2. We inserted in the manuscript the references to the two datasets available on Zenodo

3. Figure 1 contains a map that we produced using R without any copyright

4. Figure 4 has been published, as specified in the caption, on a journal with CC BY 4.0 license, therefore there is no need to ask for written permission because we follow the license requirements

We hope that you and the reviewer team like our revised paper. Please do not hesitate to contact us should any further questions arise.

Best regards,

Matteo De Felice, Derck Koolen, Konstantinos Kanellopoulos, Sebastian Busch, Andreas Zucker

---

## [Decision Letter · Decision Letter 1]

25 Jul 2023

Climate variability on Fit for 55 European power systems

PONE-D-23-00558R1

Dear Dr. De Felice,

We’re pleased to inform you that your manuscript has been judged scientifically suitable for publication and will be formally accepted for publication once it meets all outstanding technical requirements.

Kind regards,

Bruno Merk

Academic Editor

PLOS ONE

Additional Editor Comments (optional):

Congratiulations

Reviewers' comments:

Reviewer's Responses to Questions

**Comments to the Author**

1. If the authors have adequately addressed your comments raised in a previous round of review and you feel that this manuscript is now acceptable for publication, you may indicate that here to bypass the “Comments to the Author” section, enter your conflict of interest statement in the “Confidential to Editor” section, and submit your "Accept" recommendation.

Reviewer #1: All comments have been addressed

2. Is the manuscript technically sound, and do the data support the conclusions?

Reviewer #1: Yes

3. Has the statistical analysis been performed appropriately and rigorously? 

Reviewer #1: Yes

4. Have the authors made all data underlying the findings in their manuscript fully available?

Reviewer #1: Yes

5. Is the manuscript presented in an intelligible fashion and written in standard English?

Reviewer #1: Yes

6. Review Comments to the Author

Reviewer #1: The authors have addressed all comments made previously, and correspondingly revised their paper. I have no further comments.

7. PLOS authors have the option to publish the peer review history of their article (what does this mean?). If published, this will include your full peer review and any attached files.

Reviewer #1: No

---

## [Editor Report · Acceptance letter]

16 Aug 2023

PONE-D-23-00558R1 

Climate variability on Fit for 55 European power systems 

Dear Dr. De Felice:

I'm pleased to inform you that your manuscript has been deemed suitable for publication in PLOS ONE. Congratulations! Your manuscript is now with our production department. 

Kind regards, 

on behalf of

Prof. Dr. Bruno Merk 

Academic Editor

PLOS ONE